# CyVerse Austria—A Local, Collaborative Cyberinfrastructure

**Konrad Lang** [1,†] , **Sarah Stryeck** [1,*,†] , **David Bodruzic** [2] , **Manfred Stepponat** [3] ,
**Slave Trajanoski** [4] , **Ursula Winkler** [2] **and Stefanie Lindstaedt** [1,5,*]

1   Institute for Interactive Systems and Data Science, Graz University of Technology, 8010 Graz, Austria;
    konrad.lang@tugraz.at
2   Server and Storage Systems, University of Graz, 8010 Graz, Austria; david.bodruzic@uni-graz.at (D.B.);
    ursula.winkler@uni-graz.at (U.W.)
3   Central Information Technology, Graz University of Technology, 8010 Graz, Austria; stepponat@tugraz.at
4   Core Facility Computational Bioanalytics, Medical University of Graz, 8010 Graz, Austria;
    slave.trajanoski@medunigraz.at
5   Know-Center GmbH, 8010 Graz, Austria
*   Correspondence: sarah.stryeck@tugraz.at (S.S.); slind@know-center.at (S.L.); Tel.: +43-316-873-30677 (S.S.);
    +43-316-873-30600 (S.L.)
†   These authors contributed equally to this work.

**Abstract:** Life sciences (LS) are advanced in research data management, since LS have established disciplinary tools for data archiving as well as metadata standards for data reuse. However, there is a lack of tools supporting the active research process in terms of data management and data analytics. This leads to tedious and demanding work to ensure that research data before and after publication are FAIR (findable, accessible, interoperable and reusable) and that analyses are reproducible. The initiative CyVerse US from the University of Arizona, US, supports all processes from data generation, management, sharing and collaboration to analytics. Within the presented project, we deployed an independent instance of CyVerse in Graz, Austria (CAT) in frame of the BioTechMed association. CAT helped to enhance and simplify collaborations between the three main universities in Graz. Presuming steps were (i) creating a distributed computational and data management architecture (iRODS-based), (ii) identifying and incorporating relevant data from researchers in LS and (iii) identifying and hosting relevant tools, including analytics software to ensure reproducible analytics using Docker technology for the researchers taking part in the initiative. This initiative supports research-related processes, including data management and analytics for LS researchers. It also holds the potential to serve other disciplines and provides potential for Austrian universities to integrate their infrastructure in the European Open Science Cloud.

**Keywords:** computational infrastructure; Docker; bioinformatics; life sciences; HPC; computing; container; singularity

## 1. Introduction

Over recent years, increasing numbers of gatekeepers such as funding organizations (the European Commission, but also on the national level, French ANR [1], US NIH [2], UK Wellcome Trust [3], and the Austrian FWF [4]) and journals demand data sharing, data management and data stewardship according to FAIR (Findability, Accessibility, Interoperability and Reusability) principles in order to ensure transparency, reproducibility and reusability of research [5–7]. By meeting those requirements, the implementation of FAIR principles provides advantages for different stakeholders such as (i) researchers receiving credit for their work and benefitting from shared data by other researchers,

(ii) funding agencies aiming for long-term data stewardship, (iii) professional data publishers getting credit for their software, tools and services for data handling and (iv) data science communities for exploratory analyses. Making data FAIR requires skills and guidance [5–7].

Although FAIR principles are generic and not discipline-specific, the tasks to meet the requirements are in reference [8]. There are a few disciplines which develop FAIR practices and tools in an incipient stage. However, other disciplines such as life sciences (LS) already have established tools for long-term storage (e.g., https://www.ncbi.nlm.nih.gov/, https://www.ebi.ac.uk/) and standards for FAIR practice (e.g., http://www.dcc.ac.uk/resources/metadata-standards/abcd-access-biological-collection-data).

Although LS are quite advanced in developing disciplinary tools for data archiving and established metadata standards for data reuse, we detected that there is a lack of tools supporting the active research process. This leads to tedious and demanding work to ensure that research data after publication are FAIR and that analyses are reproducible. In an attempt to overcome the problem, we identified a US initiative from the University of Arizona called CyVerse US, which supports these processes from data generation, management, sharing and collaboration to analytics. CyVerse US was originally created by the National Science Foundation in 2008 under the name iPlant Collaborative. From its inception, iPlant quickly grew into a mature organization providing powerful resources and offering scientific and technical support services to researchers nationally and internationally. In 2015, iPlant was rebranded to CyVerse US to emphasise an expanded mission to serve all LS.

Merely registering Austrian researchers on a US platform to enable collaborations, data sharing and analytics is not reasonably practicable, due to GDPR restrictions in Europe for user access management, general data sharing regulations of researchers at Austrian institutions [9,10] and required NSF funding for access to CyVerse US to name a few reasons. In addition, data intensive workload for HPC using commercial cloud providers can be expensive due to data storage and transfer charges [11,12]. Given that fact, it seemed quite natural to deploy a similar, local platform for LS researchers in Graz, Austria, based on the local requirements and using institutional infrastructure. CyVerse Austria (CAT named hereafter, https://cyverse.tugraz.at) is an extensible platform deployed within the frame of BioTechMed Graz in order to support LS researchers in Austria at Graz University of Technology (TUG), University of Graz (KFUG) and Medical University of Graz (MUG) in data management and complex bioinformatic analyses using high performance computing (HPC) and supporting containerisation with Docker and Singularity.

Cyberinfrastructure (also known as CI or computational infrastructure) [13] provides solutions to the challenges of large-scale computational science. Analogous to physical infrastructures such as laboratories making it possible to collect data, the hardware, software, and people that comprise it, cyberinfrastructures make it possible to store, share and analyse data. In order to ensure reproducible research and data analytics and to get around the dependency hell (referring to frustration about software depending on specific versions of other software packages) [14], CAT is based on Docker [15] technology. Using cyberinfrastructure, teams of researchers can attempt to answer questions that previously were unapproachable as the computational requirements were too large or too complex. Moreover, collaborations are highly strengthened due to federated storage of research data from collaborating institutions. Finally, CAT enables FAIR data practices. Through adding corresponding metadata to each dataset, documentation of research data is ensured. Metadata are searchable and connected to the dataset, therefore, data are findable and accessible, meaning that users can either search their own (meta)data and/or all the (meta)data collaborators shared with them. In addition, CAT endorses and recommends the usage of international standards for data and metadata to make them interoperable. In CAT, interoperability is supported through making available standardised metadata templates. Metadata standards in CAT include common formats such as Dublin Core, Minimum Information for a Eukaryotic Genome Sequence (MIGS), Minimum Information for a Metagenomic Sequence (MIMS), NCBI BioProject, NCBI BioSample and Legume Federation. CAT users can decide whether they make use of the metadata standard templates or if they establish their own structure for metadata files. Data sharing, as well as adequate documentation, makes research data understandable,

and hence, reusable. Apart from datasets, analytics tools are saved as Docker images in CAT, ensuring the accessibility, interoperability and reusability of code [16].

The infrastructure of CAT includes (i) a data storage facility with the possibility to integrate existing storage facilities, (ii) an interactive, web-based, analytical platform for Docker containers, (iii) web authentication and security services to allow the usage of existing authentication solutions, and (iv) support for scaling computational algorithms to run on HPCs, also by using existing HPC resources.

Summarising this project helped to enhance and simplify collaborations between universities in Graz. It (i) created a distributed computational and data management architecture, (ii) identified and incorporated relevant data from researchers in LS, and (iii) identified and hosted relevant tools, including analytics software to ensure reproducible analytics using Docker technology for the researchers taking part in the initiative. In addition, it holds potential to serve other institutions as well as other disciplines. At this stage, the difference between CAT and CyVerse US lies in the management of storage and the available HPC resources. CAT offers distributed storage at the participating universities, whereas CyVerse US provides federated storage for all its users. CAT also utilises available HPC resources for affiliated universities by transferring and translating analysis jobs to this HPCs together with a core HPC available to all researchers. In addition, CAT is not accessible from outside the university network which is essential to align with university regulations on research data in Graz. Moreover, non-local user authentication is one of the next steps in the development of CAT. In contrast to that, CyVerse US is accessible for everyone but usage will be depending on ongoing NSF funding. Currently, CAT offers a limited number of modules (Data Store, Discovery Environment); additional modules from CyVerse US will be implemented in the near future (e.g., Atmosphere). Finally, CAT established a connection from HTCondor to an HPC cluster with the Son of Grid Engine resource management system which provides valuable insight for the team of CyVerse US and serves as a template for future HPC connections.

Therefore, CAT is a potent platform which is useful for researchers not only in the LS. It covers user requirements by (i) increasing efficiency in data management and sharing of data and tools, (ii) complying with funders' requirements, (iii) scaling up analyses, (iv) ensuring reproducibility when re-running analyses, and (v) creating a network of researchers with different specialities.

With this publication, we would like to present the services CAT provides for researchers, introduce the available infrastructure in Graz and elaborate on the value it adds for researchers with short use case descriptions. Therefore, we are addressing with this publication on the one hand, the HPC community to provide knowledge about state-of-the-art platforms to make HPC systems easily accessible for researchers. In addition, we think that LS researchers are looking more and more for new or already established solutions for RDM and reproducible analytics because publishers and funders are asking for FAIR solutions. Therefore, we assume that it is also a useful resource for them to understand the setup for a sophisticated RDM solution; with the elaborated use cases, we help them to identify as part of the target group.

## 2. Materials and Methods

In this paper, we outline the deployment of CyVerse as the open source cyberinfrastructure used to provide LS research within the BioTechMed Graz framework, with extensive tools for cross-university collaboration. From this deployment, available services will be shown as well as an outlook for the next steps in deployment will be given. CAT is based on CyVerse US and derived from the source code available at GitHub [17], where specific implementations for CAT can be found in the CyVerse Austria GitHub Repository [18]. The platform is built using the Google Web Toolkit [19] for the user interface, with an architecture of microservices deployed on a Kubernetes cluster to facilitate scalability and maintainability.

CAT is using a microservice infrastructure and provides services from user management with Open Lightweight Directory Access Protocol (OpenLDAP) [20] and Central Authentication Service

(CAS) [21], using integrated Rule-oriented Data System (iRODS) [22] for data management and containerisation tools like Docker with a local Docker registry. HTCondor [23] enables analysis within the CAT network and bridges to other HPC systems to facilitate usage of available computational resources. In the user portal, CAT account management is currently done by the CAT team and user information is stored at the iRODS Zone of TUG. Interested users request an account by contacting the CAT team. However, in the future, Shibboleth [24] integration will be used to have a single sign-on system for university employees. CAT receives information about institutional affiliation of the user too. Data storage and data management are integrated in the Discovery Environment (DE), meaning that users see their data where tools for analysis are stored.

### 2.1. Data Management

Integrated Rule-Oriented Data Systems (iRODS) Deployment

The integrated rule-oriented data system (iRODS) is a central building block of CAT and enables federated data storage between different storage locations. Data discovery is facilitated by using a powerful search engine built with Elastic Search [25], which provides a rule engine to enforce actions in distinct locations, a global namespace and a metadata catalogue. Metadata in iRODS are organised as Attribute-Value-Unit (AVU) triples and can be configured to reflect any ontology to describe the stored metadata. Thus, iRODS builds the fundament for data management, organisation, storage and sharing in CAT (Figure 1).

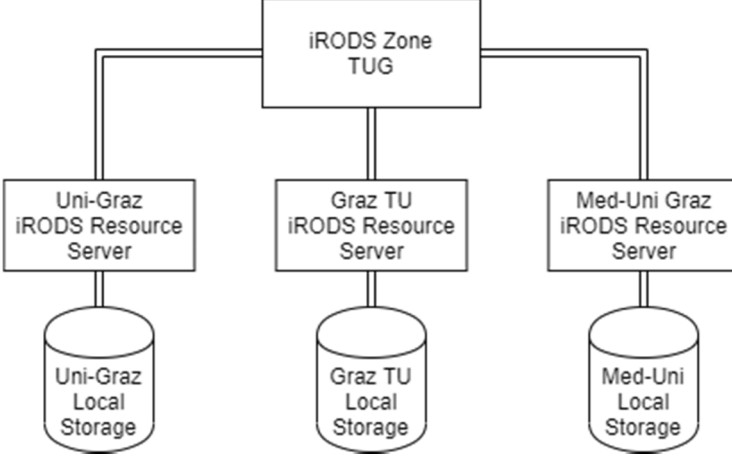

**Figure 1.** iRODS Deployment.

The iRODS deployment for CAT consists of one iRODS catalogue server (iCAT) holding the metadata for the zone TUG. The iCAT server is located in the central deployment at the CAT core, and manages metadata for all available iRODS resources servers at KFUG, MUG and TUG. For this deployment, no replication or distribution of data is allowed, as data should not leave the universities' boundaries unintentionally. Nevertheless, each iRODS resource server can be configured to allow data transfer to a trusted HPC cluster to deliver data required for computations to a remote location that is outside the boundaries of the specified universities.

### 2.2. Analytics

2.2.1. Discovery Environment

On top of iRODS, this middleware platform is complemented with the CyVerse Discovery Environment (DE). DE enables the upload of analytics tools, packed as Docker images, in order to perform analytics operations. Hereby, a bioinformatics tool together with all its dependencies is

wrapped into a Docker image to run in a reproducible manner regardless of the software/operating system environment. Moreover, multiple versions of a given bioinformatics tool can be stored in DE, which ensures that a data analysis performed five years ago with a given version can be repeated in CAT. Those tools can schedule jobs to HPC clusters, manage analytics output and notify users about the status of a computation. DE communicates with other components using a Representational State Transfer (ful) Application Programming Interface (REST-ful API). These apps can be provided by individual users of CAT, and then, are made available to researchers.

The code for DE of CyVerse is available on GitHub [26] with specific implementations for CAT to be found in reference [27].

In combination, iRODS and DE provide a service architecture which enables researchers to manage, store and share data, but also computational processes, products of workflows and analyses (Figure 2).

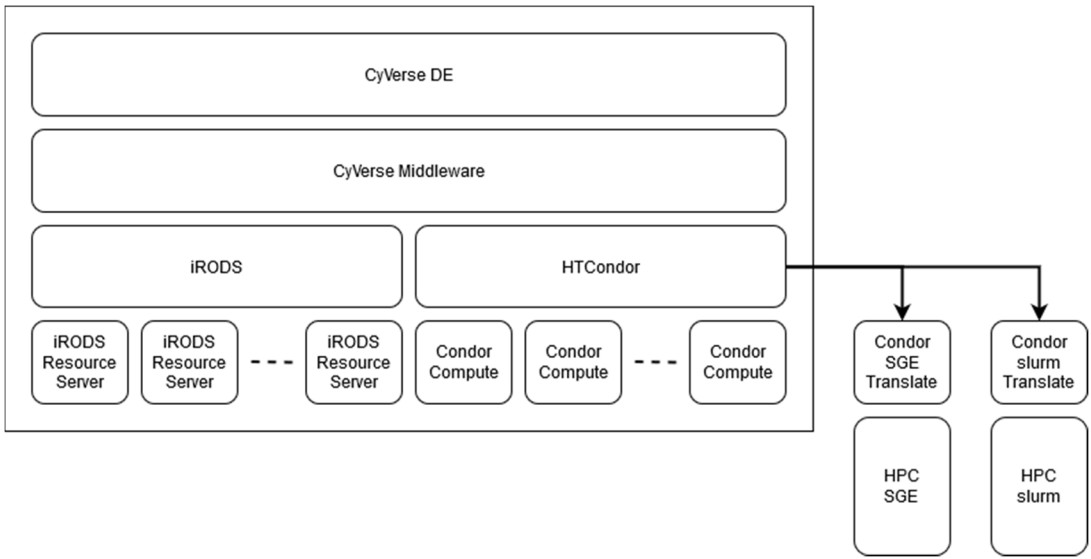

**Figure 2.** CAT discovery environment HPC stack.

### 2.2.2. HPC Connection

The computations on the data are handled by CAT utilising HTCondor to manage the available computational resources. Analysis tasks are containerised using Docker technology (Apps in DE), and verified Docker images are stored in a private Docker repository. The implementation of CAT uses Docker also to package the data staging in and out, allowing the containerised app and the contents to be transferred to any available computational resource.

Additional development was necessary to connect to other HPC resources outside of CAT core deployment. Their utilisation requires supplemental information about affiliation and access for individual users in order to send analysis apps to connected HPC resources. The extra development work done makes use of the "grid" feature provided by HTCondor and sends jobs with matching ClassAds to provisioned HPCs.

### 3. Results

CAT offers users two layers for (i) research data management and (ii) reproducible analytics using Docker technology. While CyVerse US has grown out of iPlant, the principles of data intense research and the underlying supportive patterns can be applied to all sciences with comparable demands. From the experiences of CyVerse US and the implementation of a cross-institutional research cyberinfrastructure [28–30], we concluded that to perform the deployment of CyVerse as an Austrian cyberinfrastructure will have the most benefit to our researchers. CAT serves the community as a powerful infrastructure, providing data storage, data management, data sharing and management of

HPC resources and APIs. Furthermore, as extra benefits, it enables the usage of existing storage solutions and utilisation of available HPC clusters to perform data analytics using tools in Docker containers.

*3.1. Data Management*

### 3.1.1. Integrated Rule-Oriented Data Systems (iRODS) Deployment

The CAT data management provided is based on the iRODS technology stack, which allows users to maintain, share and analyse data conveniently. Furthermore, the storage can be distributed geographically. Using iRODS rules data can be managed in a way to adhere to specific rules for each of the participating universities. Besides requiring data to be uploaded into CAT for management and analysis, iRODS also provides metadata-only datasets for store requests. In this case, iRODS is configured to perform periodical scans of accessible data storage, and add, delete or update information about available data. This information is reflected in the DE, and for a researcher, the location of the actual data is transparent. With this scanning, researchers can easily add large datasets and use them for analysis within CAT. Though the benefits outbalance the disadvantages by far, there are some cases in which, e.g., data can change or be removed without being reflected in the DE, or data needs to be transferred through networks which are not designed for high-throughput analysis.

iRODS holds potential to be integrated in much bigger environments such as the European Open Science Cloud (EOSC) [31]. It is also the technology behind EUDAT safe replication B2SAFE [32]. B2SAFE is one of the EOSC storage services, which is robust and safe and allows community and departmental repositories to implement data management policies on their research data in a trustworthy manner [33].

In our setup, we connected three different universities in Graz. All of them had a different initial situation and infrastructure and serve researchers from different disciplines. In this project, they were connected to establish a joint network for the local community, demonstrating the versatility of CAT to combine heterogeneous institutions to provide equal functionalities for all of them. In the following, we will describe the three different partners.

### TUG Main Node

The main node at TUG is providing the DE, Data Store and the HTCondor analytics layers to all of its users.

TUG has divided its research into five innovative areas, so called Fields of Expertise. Researchers in the Fields of Expertise break new ground in basic research. The following Fields of Expertise do exist: (i) Advanced Materials Science, (ii) Human and Biotechnology, (iii) Information, Communication and Computing, (iv) Mobility and Production and (v) Sustainable Systems. In order to serve those fields of expertise, Graz University of Technology offers IT services for employees and researchers. Services for researchers include access to European research networks (e.g., ACOnet, GÉANT, PRACE), server hosting, housing, but also support in (i) high performance computing, (ii) cloud storage, (iii) project management tools, (iv) software for research and with the CAT project, also (v) data analysis. In (i), computational resources are provided on local IT systems (Ceph [34]/Galera/Linux cluster, GPU) and connection to the Vienna Scientific Cluster (VSC) [35] for computer-aided scientific work with high resource requirements. In (ii), a private cloud supports researchers and their collaborations with modern technical capabilities. In (iii) and (iv), researchers get access to specific software using agile and classic project management methods and also technical academic software with campus or site licence for academic research. Finally, with (v), TUG supports researchers by providing CAT [36]. By integrating CAT in the central services of the IT department of TUG, this platform serves the target group, LS. However, it has the potential to serve all researchers at the institution, which is also the future direction of CAT.

KFUG Node

Some years ago, the KFUG had to react to the challenge of the increasing computational demands of LS which could not be covered with the usual "traditional" HPC Clusters, i.e., architectures with one (several) master node(s) and many slim slave nodes—slim in terms of the Central Processing Unit (CPU), Random Access Memory (RAM) and local diskspace resources, but strongly connected to each other over high speed networks and sharing fast distributed filesystems. This architecture concept had been well adapted, especially to Message Passing Interface (MPI) or comparable codes running simultaneously on many nodes' distributed processes. LS of the KFUG with codes much heavier in I/O but less intense CPU demands had, in the meanwhile, to look for separate and mainly own local specialised hardware, the upcoming cloud-devices or the infrastructure of project related cooperation partners.

In reaction to finding a collective solution for these growing field of fairly wide-scattered applications as well as some fraction of the established HPC community, the idea of building up a loosely connected cluster of fat nodes (i.e., in CPUs, RAM, local diskspace) and (as the budget was moderate) neglect the high speed network component, but still using the proven configuration of a master node, processing nodes and several distributed filesystems useable by codes via a job scheduler seemed to be the best solution. To codes not well suited for schedulers, the possibility to reserve one or several nodes is offered as an option. The concept worked very well from the start in 2014 until now; meanwhile, the cluster has been upgraded to actually 88 nodes of different hardware architectures. The next step came in 2017 with the start of the CAT project. Integrating the cluster into this cooperation project of several universities appeared quite natural, as the cluster is well suited for CAT Batch Processing. Of course, first there were some problems to solve such as (i) CAT uses another type of job scheduler (HTCondor) and the cluster works with the Son of Grid Engine (SGE) [37], so there was the need of programming an interface, or (ii) difficulties with the conversion of Docker apps from CAT to Singularity [38] on the HPC cluster. Both incompatibilities could be managed, and meanwhile, CAT batch jobs are fully integrated without any need of an additional queue or other special configurations.

MUG Node

MUG is a young university and has been autonomous since January 1st, 2004 from KFUG. Until recently, HW infrastructure for data analysis was mainly present as single island solutions at research institutes with no possibility for data sharing or providing calculation services outside the given institute. In 2018, new HPC infrastructure (MedBioNode) [39], with constant support from the IT department of the MUG, was implemented, driven by an initiative of the bioinformatics team from the Core Facility Computational Bioanalytics at the Center for Medical Research. During the planning phase, the strategic decision was taken to use CEPH [34] high performance distributed storage as the central data sharing space. Thereby, we were able to provide a modular system with an easy upgrade path in computing power and storage space. Slurm [40] was selected for cluster management.

The main aim in this initiative was to build a system that would be easy and efficient to use by researchers with different levels of knowledge in computational biology. To fulfil these requirements, Galaxy [41,42] web-based platform was implemented on top of the HPC. With its graphical user interface (GUI) and no need to install any software on the local computer, but still having access to thousands of different data analysis tools and running them on high performance infrastructure, Galaxy provides the opportunity to learn and start individual analyses in a short time. Nevertheless, more proficient cluster use is possible via command line, where the implemented open source package management system and environment management system CONDA [43] gives access to all data analysis tools.

With the achievement of centralising data storage and data analysis, it was possible to go for the next step and connect this infrastructure with the other two universities in Graz, KFUG and TUG, using CAT. This enables more efficient collaboration outside MUG and easier project coordination for

multicentre and multipartner grants. CAT jobs can be executed on the MedUni HPC (MedBioNode), where CAT ensures that sensitive data remains inside the home university.

Summarising

HPC cluster integration also has shown that CAT has the property of being scalable to much bigger dimensions without too much effort, as most development work has already been done. The only remaining deficiency concerns the concept of a collective user-authentication process (e.g., Active Directory). However, this is not an isolated problem of CAT, and finally, it requires a common concept for all institutions to join.

### 3.2. Analytics

#### 3.2.1. HPC Connection

TUG provides one HTCondor node with 256 CPU cores, 512 GB RAM, 10 TB global scratch space for data of all CAT users utilised during analysis

The HPC cluster from KFUG with 88 cluster nodes provides 1822 CPU cores, 8 GPUs 100 TB global scratch.

MedUni HPC (MedBioNode) has 9 compute nodes, one login/master node and on Galaxy server node with total number of 284 CPU cores, 4.256 TB working memory and 4.3 PB CEPH shared storage space (Figure 3).

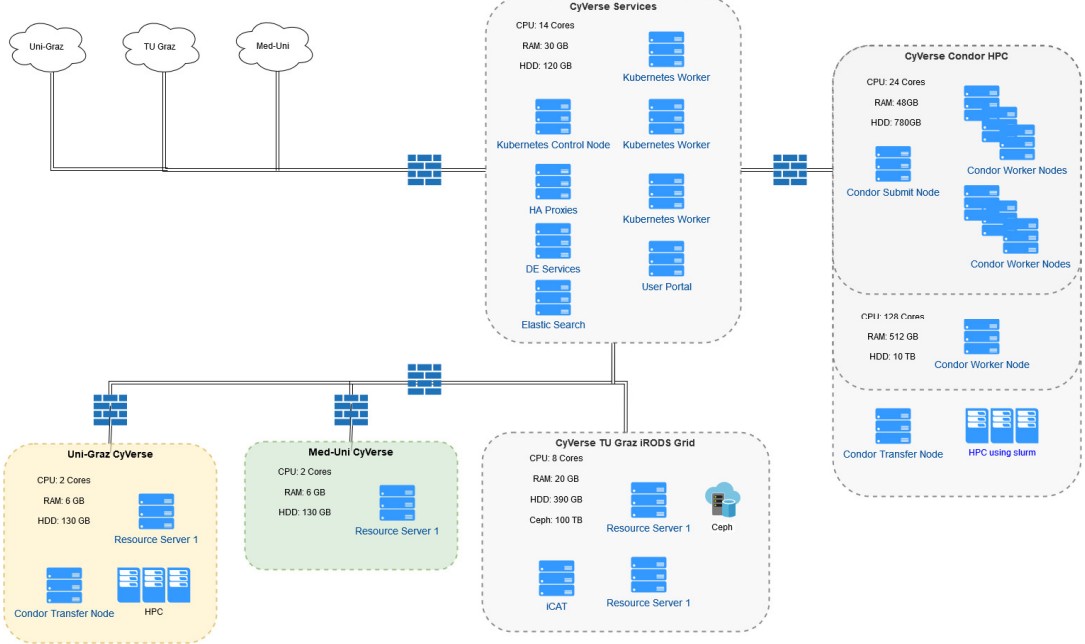

**Figure 3.** CAT Technical Deployment.

This figure shows the current deployment of CAT at the three universities in Graz.

CAT users are eligible to use the HTCondor node provided by TUG, and, based on their affiliation, may also have the possibility to use the HPC provided by KFUG to perform analysis. To support this, CAT may transfer jobs to the HPC system of KFUG. The decision to transfer jobs to another HPC is based on the affiliation using the condor-launcher sub-system, as depicted in Figure 4. For CAT jobs not transferred to the KFUG HPC, the submission of an analysis uses Docker containers. A single job run is controlled by HTCondor and submitted to the main HPC node cy-node01. There, the analysis is wrapped in 3 distinct steps: (1) connecting to the data store using iRODS to provide the data to the

analysis container, (2) the analysis container where the actual calculations take place, and (3) connecting to iRODS to put the data from the analysis in step 2. To have continuous updates on the job status, the Advanced Message Queuing Protocol (AMQP) [44] is used. Whenever a user has the affiliation and rights to use HPC resources at KFUG, jobs are managed differently. These external jobs can be run using other HPC resources managed by other queueing systems as SGE [37] or slurm [40]. In the following, the job is run using SGE scheduler on the KFUG HPC system as described in detail, but this can also be applied to systems using slurm as the workload manager. Such analyses are performed by transferring the job description via a secure VPN connection to an HTCondor transfer node located at KFUG. At the transfer node, the job description is translated into a job description for SGE. As the KFUG cluster only provides Singularity, additional steps are required to transform the Docker container to a Singularity container. Data and status updates are transferred using secured VPN connections to provide CAT with information using AMQP. Data transfer itself is managed by iRODS and ensures all data required to run the analysis and results are available at the HPC as well as in CAT.

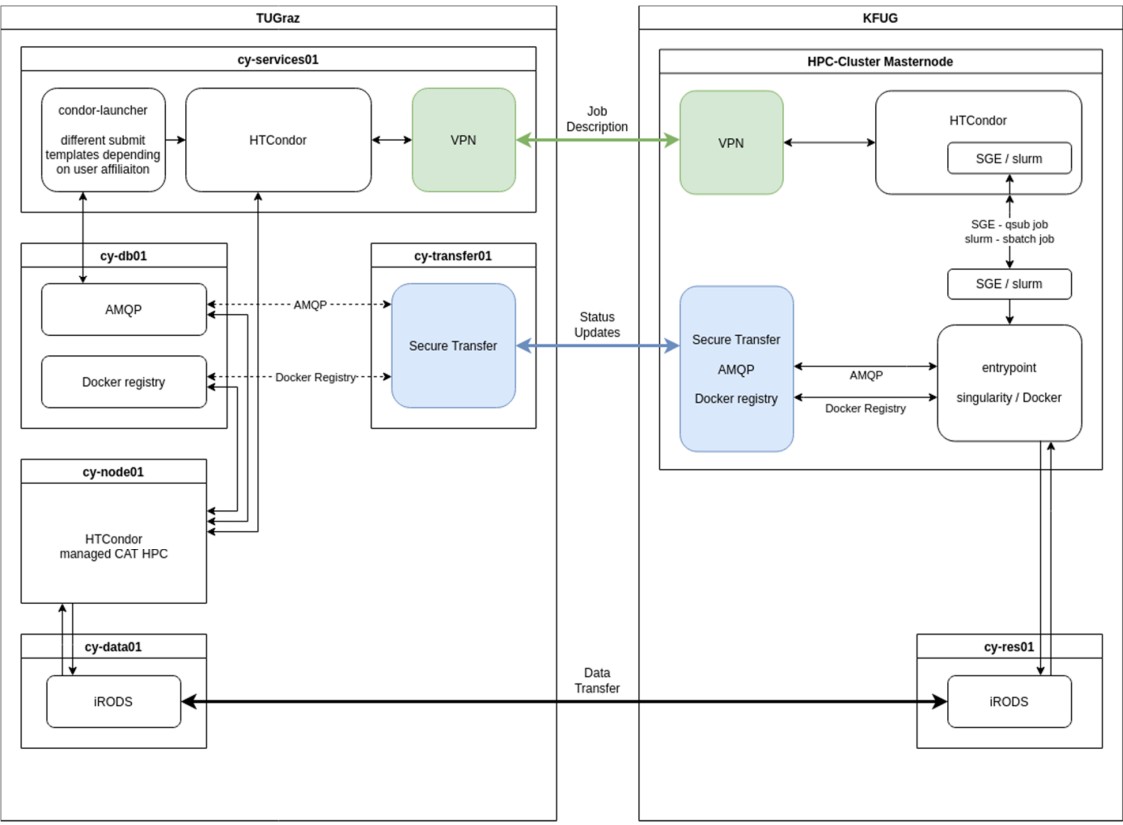

**Figure 4.** Job routing utilising HTCondor.

### 3.2.2. DE and Docker

Sometimes, the repeated analysis of the same dataset does not result in the exact same result. Reasons include the lack of documentation of data analysis or a new software version with minimally changed parameters. Last but not least, researchers complain about the so-called dependency hell.

The solution to ensure reproducibility of analyses is container technology. There are different types of containers. As already elaborated in 2.2.1. Discovery Environment, CAT makes use of Docker containers in the DE. Analytics in the DE can be performed as scripts running in the background. However, it is also possible to use the Visual and Interactive Computing Environment (VICE) for the interactive use of applications such as Jupyter notebooks [45], RStudio [46] and RShiny [47]. Those VICE applications run on an open port for enabling their web user interface. Here, it is important to mention that notebook science is important to ensure reproducibility of analyses, which has been shown by

several works [48,49]. For instance, Jupyter Notebooks are open source web applications allowing users to create and share documents that contain live code, equations, visualisations and narrative text. By combining all this information, input/output data as well as the code for the underlying analyses are FAIR [45].

Some limitations concern the containerisation of applications, such as limited access to large data or no web user interface. Those shortcomings are tackled by CAT DE by providing a web interface for streamlining data upload, organisation, editing, viewing, sharing as well as task execution. As already mentioned, HTCondor is used to schedule jobs to the HPC clusters. In the DE, the Docker engine runs on three different containers: (i) one for data delivery, (ii) one with the specific dockerised tool and (iii) one for returning data to the Data Store.

The integration of new tools as Docker images into CAT is done easily via providing a link through the GUI to lower down the barrier for the individual researcher. Through the 'request tool' button, users can submit the Dockerfile by either entering the link, uploading it or selecting an existing file. A detailed description can be found in reference [50].

*3.3. Users*

With this publication, we would like to present the technical deployment of CAT, and, in addition, we aim at emphasising the importance of CAT as a tool for researchers' data management and analytics. We will elaborate on (i) what CAT offers to researchers, (ii) who should use CAT, and (iii) our experiences/lessons learned after four years of project duration.

3.3.1. What Does CAT Offer to Researchers?

CAT is a so-called cyberinfrastructure, meaning that it provides solutions to challenges of large-scale computational science. It includes hardware and software which, in combination, make up innovative solutions for users for data management and analytics, as described below:

- Data management: (i) Data transfer: CAT data transfer is enabled by the third-party software tool Cyberduck [51] to drag and drop files between your computer and the Data store. In addition, iCommands, a collection of tools developed by iRODS, enables command line based data transfer. (ii) Data storage in the Data store: data can be managed in file systems comparable to a Windows explorer system with folders and subfolders. (iii) Metadata: In order to make data FAIR, one can add metadata to a single file or in bulk to large collections of data in the Data store. (iv) Data Sharing: Research data deposited in CAT can be shared with other CAT users. Users can assign read, write or own rights to the data [52].
- Data analytics: The DE offers a variety of tools for bioinformatics analyses to run as a script in the background, but also as a VICE app, with web-based user interface; check whether your tool is already available in the DE (CAT or others—transfer from CyVerse US tools is straightforward). You can find a more detailed description in reference [50]. In order to implement a new tool of interest to CAT, you need to (i) install Docker and all its dependencies on your local machine and (ii) have a URL for the tool and all its executables from a reliable source (e.g., GitHub).

3.3.2. Who Should Use CAT?

Basically, CAT is a suitable platform for any researcher in LS, but is extendable to various other disciplines since it supports standardisable research processes like data management and analytics. With our user community, we had many discussions and collected benefits that CAT brings for researchers. The following statements indicate for which problems CAT builds a potent solution.

- *'I would like to increase the efficiency and effectiveness of my day to day work with simplification of data management and publication processes.'*
- *'I want to apply for a European Commission call and I need a data management solution that is EOSC conform.'*

- *'I would like to connect my research data with appropriate standardized metadata to ensure adequate documentation of my data.'*
- *'I am collaborating with researchers at other institutions and we need a way to safely transfer data.'*
- *'I am working in a collaborative project with different research institutions and therefore different storage locations. However, we would like to have a combined data management, where we can see and structure all data, even if they are not stored at the same place.'*
- *'I would like to ensure reproducibility of my bioinformatics analyses also few years later (e.g., preserve analytics tool).'*
- *'I would like to automatize monotonous steps in my daily work, such as metadata extraction.'*
- *'I would like to create an easy-to-use, intuitive and standardized workflow for data analytics of command line-based applications for my PhD students.'*
- *'I would like to work with a platform that uses existing standards for data semantics and community standard practices for data management in order to ensure interoperability.'*
- *'I would like to use the HPC cluster at our university for computation and I need a safe way to transfer my data.'*
- *'I would like to keep up motivation and productivity of my PhD students by reducing time dedicated to administrative data management processes.'*
- *'I would like to stay up-to-date with the ongoing research in my local network in Graz.'*
- *'I would like to become part of a global network with a strong discipline-specific community developing tools for state-of-the-art research.'*

### 3.3.3. CAT Use Cases

Statements in Section 3.3.2. indicate which problems can be tackled using a CI such as CAT. In the following, we briefly introduce some of the use cases currently running in CAT. Currently, we have around 20 researchers with a CAT account working on their independent projects. Users have permanent access to CAT, downtimes are kept low for development tasks, however, in the future, a test system for CAT will be introduced which will further reduce any downtimes. With this section, we would like to further introduce problems solved by CAT, by means of practical examples.

- Use Case 1: A researcher in structural biology has an analysis pipeline which needs several command line-based tools for protein structure prediction. The output of the first tool feeds as input file into the next tool and so on. In order to ensure that tools are properly combined and correct input files are chosen by different users in the lab, creation of an automated workflow to combine the analysis is crucial to ensure reproducibility. CAT was used to import all pieces of the analysis pipeline as docker images and a workflow was defined. Now, the whole team can use the workflow in a reproducible manner.
- Use Case 2: A researcher in computational bioinformatics is teaching a course with 20 students. To ensure that all students work with a shared dataset and regardless of their own hardware, with the same analytics tools and without having any problems with underlying dependencies, he chose CAT. All students receive a CAT account and do their analyses on the same platform using the data shared on the same platform.
- Use Case 3: A researcher in biology performs genomic data analysis requiring high computational power. Since institutes usually cannot serve such requirements, he uses the institutional HPC cluster. Therefore, he has to send his locally stored data to the HPC cluster and to ensure that his tool is available as a Singularity container for analysis on the HPC cluster. By using CAT, he stores his data and analysis tools on the platform which has a connection to the HPC cluster. Thereby, data and tools will be directly transferred and output is stored again in CAT.
- Use Case 4: A researcher from KFUG is leading a project with a consortium including TUG and MUG. As a coordinator of the project, the researcher has to ensure that data management of the whole project is done according to FAIR principles. Therefore, it is essential for them

to have a central platform where all research data from the project is findable. By using CAT, they ensured that all data are either stored on the CAT storage resources or registered in iRODS making data findable.

In the time of the COVID19 pandemic, digital tools for research became even more important. With CAT, we enabled our researchers to have access to their research data and to perform all their analyses independent of the underlying hardware they were using. Currently, our group of users consists mainly of researchers from the field of LS. However, we are planning to expand and to include also other disciplines.

With CAT, we introduced a new local, collaborative CI in Graz, Austria. The technical deployment is complete, and the first researchers are using CAT. Users are supported by the CAT core team, consisting of technical experts as well as a data steward with a background in LS. Due to the incipient stage of CAT, extensive evaluation of the CI from a scientific standpoint will be done in future. However, due to experiences from CyVerse US, this platform has already proven its utility for serving LS researchers. Nevertheless, specific requirements from our local community will be evaluated in the near future.

### 3.3.4. Experiences/Lessons Learned

The project for the deployment of CAT in Graz started in 2016. We learned that much more is needed than simply providing the technical infrastructure at universities. Now, four years later, we can look back and break it down into the following lessons learned:

- Importance of support from CyVerse US. CAT is a local, independent instance of CyVerse; however, it is part of a global community with a lot of experience and know-how. The CAT team is steadily exchanging knowledge with the team from CyVerse US. There are a lot of joint development tasks in specific areas (e.g., HPC connections) ongoing. Moreover, tools for data analysis are easily exchanged between CAT and CyVerse US, showing the international level of collaborative science and the importance of FAIR.

- Data protection is a very important topic, mainly in the contexts of Medical Universities and industry collaborations. During the deployment of CAT and the connection of the MUG to our network, we learned that research data sharing, mainly in the clinical context, is an extremely sensitive topic. This leads to the necessity of tight regulations to ensure that it is clear which data can be shared in which format.

- Docker containerisation is a very nice, convenient and innovative tool for light conservation of data analytics software. However, due to superuser privileges that come with Docker containers, it is essential to use Docker-to-Singularity conversion to be able to work with the tools on HPC clusters.

- Although new hardware is permanently acquired for CAT, storage capacity and computing power will always remain a main limiting factor for researchers due to the accelerating progress in the development of big data disciplines, and therefore, also the need for high computational power. Therefore, the access to shared HPC resources is essential. With this, institutions can ensure that enough resources are available for researchers. Researchers can scale up their analyses if needed, but, in case they do not need the resources, computing power is available for other users.

- Motivation to invest time in learning the new technologies offered in CAT is essential. CAT offers many advantages for researchers; however, it requires some effort to learn how to use the platform and to reorganise existing data to the CAT Data Store. Hence, a certain commitment and motivation on the part of researchers is needed. Furthermore, the crucial aspect on maintaining user motivation is the investment in social aspects like build-up and care of a community and training possibilities. It is not sufficient just to provide the technical infrastructure; the human aspect has to be supported too. In addition, know-how for user support in CAT is required. CAT does not only imply new technologies for researchers, it does this for the IT staff too. It is

of vital importance that there are adequate training possibilities offered to ensure a competent support staff keeping up-to-date with the fast-evolving progress of this field.

## 4. Discussion

This initiative introduced locally in Graz for the BioTechMed universities a platform which ensures easy and straightforward data sharing, and thereby, ensuring fruitful collaborations. In addition, it supports handling of research data according to FAIR principles by providing a secure distributed data management system with metadata standards.

### 4.1. Benefits of CAT for LS and HPC

Taken together, regarding our experiences from the last few years, we could observe different benefits that CAT brought for LS researchers and for the HPC community. On the one hand, CAT disconnects HPC know-how for users, meaning that LS researchers without an extensive know-how about how to connect to HPCs can easily do their calculations on resources available via CAT, thereby simplifying and speeding up the research process. In addition, HPCs move more and more in the direction of big data, and, big data researchers need more access to HPC resources due to the required computational power. By combining the researchers with the HPC resources, both fields benefit through either simplified research processes (LS) or extending their user community (HPC).

### 4.2. Future Direction of CAT

Due to the very general design of the platform, DE not only has the potential to serve life scientists, but also diverse other disciplines without having to reengineer the entire system. Apart from this aspect, scaling up the existing platform opens up possibilities to expand the infrastructure to more Austrian universities and/or also to other countries. Therefore, one of the next steps could be the connection of national HPC clusters, e.g., the Vienna Scientific Cluster. Another very important issue is an easy way of data sharing for collaborations with external universities/institutions/partners by providing access using iRODS to upload, download and manage data without the need of being fully integrated in the CyVerse landscape.

For the future, computer science and data science experts as well as domain experts need to develop a collaborative strategy to ensure and streamline usage of platforms such as CyVerse. Those approaches are becoming even more important with the continuing increase of interdisciplinary research.

Furthermore, there is a large global community working on CyVerse and a lot of development is going on. All the functionalities of CyVerse are adjusted to user requirements in the course of time. This is also the reason why a new version of DE is currently under development and will be released by the end of this year. The new version of the CyVerse DE is providing a responsive and modern design, in contrast to the GWT based current version [19].

**Author Contributions:** D.B.: Methodology, Software, Writing—review and editing; K.L.: Methodology, Software, Writing—original draft, Writing—review and editing; M.S.: Methodology, Software, Writing—review and editing; S.L.: Conceptualization, Supervision, Writing—original draft, Writing—review and editing, Funding acquisition; S.S.: Conceptualization, Methodology, Project administration, Writing—original draft, Writing—review and editing; S.T.: Methodology, Software, Writing—review and editing; U.W.: Methodology, Writing—original draft, Writing—review and editing. All authors have read and agreed to the published version of the manuscript.

**Funding:** This research was funded by the Austrian infrastructure program 2016/2017, Bundesministerium für Bildung, Wissenschaft und Forschung Austria, BioTechMed/Graz Hochschulraum-Strukturmittel 'Integriertes Datenmanagement'. The project was supported by the Strategic Project Digitale TU Graz (Graz University of Technology). The APC was funded by Austrian HPC conference.

**Acknowledgments:** The authors wish to thank Simon Kainz (ZID, TUG), Christian Marzluf and Guenther Berthold (ZID, KFUG) for hosting technical infrastructure. Open Access Funding by the Graz University of Technology.

**Conflicts of Interest:** The authors declare no conflict of interest. The funders had no role in the design of the study; in the collection, analyses, or interpretation of data; in the writing of the manuscript, or in the decision to publish the results.

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
