# Peer review of "CyVerse Austria—A Local, Collaborative Cyberinfrastructure"

_mca, doi:10.3390/mca25020038_

Round 1

Reviewer 1 Report

This is an implementation paper that describes the experience of replicating an existing CI.  While there is merit in implementing CI, the authors missed a great opportunity to demonstrate how the CI is enabling research through real use cases and concrete examples of how and in what specific examples the different components are achieving reproducibility, data accessibility, etc (FAIR) principles. This is an overview, a presentation with too many HPC terms, but little about how the CI is currently performing. Who uses it, for what, when and how? How is it managed from a scientific standpoint, and how is it evaluated? The phrases from researchers at the end of the paper are requirements that could have been used in the introduction and later as pointers to demonstrate how the infrastructure is really enabling their work through use cases. Narratives about how the platform is advancing both  LS and the HPC fields are lacking.  

Author Response

Response to Reviewer 1

This is an implementation paper that describes the experience of replicating an existing CI.  While there is merit in implementing CI, the authors missed a great opportunity to demonstrate how the CI is enabling research through real use cases and concrete examples of how and in what specific examples the different components are achieving reproducibility, data accessibility, etc (FAIR) principles.

Response: We thank the reviewer for his/her comment. We did not provide too much insight in the currently ongoing projects on CAT because our users are also planning to publish e.g. their workflow or data merging protocols as independent manuscripts. However, we agree that practical examples are very helpful to understand the value and versatility of CAT, therefore we included in our manuscript a new section about use cases. We would like to emphasize here that we are not able to be more concrete about our use cases for the reason mentioned above

This is an overview, a presentation with too many HPC terms, but little about how the CI is currently performing. Who uses it, for what, when and how?

Response: We included statements about usage of the CI in 3.3.3. We have currently around 20 users and, as explained above, we also included more detailed information about ongoing use cases (respecting their independent publishing intentions).

In addition, we agree that the descriptions of the three institutions and their respective HPC systems are very detailed. Our intention here was to demonstrate the versatility of CAT to in combining heterogeneous institutions to provide equal functionalities for all of them.  

How is it managed from a scientific standpoint, and how is it evaluated?

Response: We have to state (and we also did now in the manuscript) that we recently finished the technical deployment and just started our first use cases. At this stage it is not possible for us to do a detailed evaluation of CAT. However, based on experiences from CyVerse US, we already have seen the scientific value. In addition, we plan to evaluate CAT as soon as most of our use cases are completed.

The phrases from researchers at the end of the paper are requirements that could have been used in the introduction and later as pointers to demonstrate how the infrastructure is really enabling their work through use cases.

Response: We agree that including use cases is important to show the potential of CAT, we therefore included them in 3.3.3. We also included a short summary ‘Who should use CAT’ in the introduction.

Narratives about how the platform is advancing both LS and the HPC fields are lacking.  

Response: We agree that we should emphasize the benefits for LS and HPC fields more. We therefore included a section in our discussion elaborating on benefits of CAT in 4.1.

Reviewer 2 Report

Title: CyVerse Austria—A Local, Collaborative Cyberinfrastructure

The goal of this paper is to describe the deployment of CAT, a localized deployment of CyVerse in Austria.  The key needs CAT addresses are to improve scientific collaboration across Austria, respect and protect data privacy as required by the EU, and help researchers adhere to FAIR data management principles.  

Overall, this is a nice paper.  Besides form my comments below, this paper needs to be copy edited.  There are many grammar and spelling mistakes that need to be corrected before publication.  In addition, I suggest the authors review all of the software systems used and described in the manuscript.  Nearly all of them have academic references and these need to be included. 

Comments:  

  • Not sure how widely know the term “dependency hell” is on line 81.  May want to define and/or reference.
  • End of introduction:  add some discussion about how a shared cyberinfrastructure enables collaborations.  Specifically, researchers with different specialities and easily share data and analytical tools, and rerun and scale-up analyses.
  • End of introduction: Should tie back to how CAT has enabled FAIR data practices 
  • Provide a reference for iRODS and other software systems listed in the paper.
  • Line 135: “On top of iRODS, this middleware platform is complemented with the CyVerse DE” -- change to “On top of iRODS, this middleware platform is complemented with the CyVerse Discovery Environment (DE)
  • Expand upon how tool integration works -- any dockerized tool available through common platforms such as https://github.com/BioContainers can be added.  The core concept is to lower the barrier to getting a new tool integrated for use by any researcher.
  • Line 149-150: “These Apps can be provided by individual users of CAT and then are made available to researchers.” seems to fit better when describe the DE.
  • Is there any interop between CAT and CyVerse US?  E.g., exchange tools, workflows, and data?  If so, this would show the international level of collaborative science and the importance of FAIR.
  • Reference missing for Galaxy on 240
  • Section 3.2.2 (line 269) repeats information presented earlier in the manuscript
  • VICE is squeezed into line 280-282.  Notebook culture in science is and important, and should be described in more detail.  For example, how does this increase reproducibility and accessibility to analyses?  How does it fit with FAIR?
  • Line 357:  I would add that access to “shared” HPC is important.  You want researchers to be able to scale up when they have the need, and then have those compute resources available to others so others can use them. 

Author Response

Response to Reviewer 2

The goal of this paper is to describe the deployment of CAT, a localized deployment of CyVerse in Austria.  The key needs CAT addresses are to improve scientific collaboration across Austria, respect and protect data privacy as required by the EU, and help researchers adhere to FAIR data management principles.  

Overall, this is a nice paper.  Besides form my comments below, this paper needs to be copy edited.  There are many grammar and spelling mistakes that need to be corrected before publication. 

Response: We are delighted that the reviewer acknowledges the value of CAT, how we address demands for scientific collaborations and help researchers to adhere to FAIR principles. We thank the reviewer for making us aware of grammar and spelling mistakes throughout the manuscript. We want to mention that we consulted a native scientific writer for proof-reading and corrected/improved the manuscript to the best of our knowledge. In case we missed remaining mistakes, we would be very grateful if the reviewer could point us to them.

In addition, I suggest the authors review all of the software systems used and described in the manuscript.  Nearly all of them have academic references and these need to be included. 

Response: We included the corresponding references to the best of our knowledge.

Comments:  

  • Not sure how widely know the term “dependency hell” is on line 81.  May want to define and/or reference.

Response: We included a short definition as well as a reference for dependency hell.

  • End of introduction:  add some discussion about how a shared cyberinfrastructure enables collaborations.  Specifically, researchers with different specialities and easily share data and analytical tools, and rerun and scale-up analyses.

Response: We tried to elaborate more on these points by adding a section in the introduction. In addition, we also included more examples about ongoing use cases on CAT. However, we did not provide too much insight in the currently ongoing projects on CAT because our users are also planning to publish e.g. their workflow or data merging protocols as independent manuscripts.

  • End of introduction: Should tie back to how CAT has enabled FAIR data practices 

Response: We included one section on CAT and FAIR data practices in the introduction.

  • Provide a reference for iRODS and other software systems listed in the paper.

Response: We included the corresponding reference.

  • Line 135: “On top of iRODS, this middleware platform is complemented with the CyVerse DE” -- change to “On top of iRODS, this middleware platform is complemented with the CyVerse Discovery Environment (DE)

Response: We added the full name of the Discovery Environment.

  • Expand upon how tool integration works -- any dockerized tool available through common platforms such as https://github.com/BioContainers can be added.  The core concept is to lower the barrier to getting a new tool integrated for use by any researcher.

Response: We agree that it is important to mention that import of new tool is easy for researchers. We included a statement in the manuscript and referred to a publication elaborating on this topic.

  • Line 149-150: “These Apps can be provided by individual users of CAT and then are made available to researchers.” seems to fit better when describe the DE.

Response: We moved the statement to the DE description.

  • Is there any interop between CAT and CyVerse US?  E.g., exchange tools, workflows, and data?  If so, this would show the international level of collaborative science and the importance of FAIR.

Response: We included in our Experiences/Lessons Learned section also a longer description about our collaborations with CyVerse US and that this shows the international level of collaborative science.

  • Reference missing for Galaxy on 240

Response: We included the corresponding reference.

  • Section 3.2.2 (line 269) repeats information presented earlier in the manuscript

Response: We tried to condense down the section to the essential parts.

  • VICE is squeezed into line 280-282.  Notebook culture in science is and important, and should be described in more detail.  For example, how does this increase reproducibility and accessibility to analyses?  How does it fit with FAIR?

Response: We included a more detailed statement on notebook culture in science in the manuscript.

  • Line 357:  I would add that access to “shared” HPC is important.  You want researchers to be able to scale up when they have the need, and then have those compute resources available to others so others can use them. 

Response: We included a statement on shared HPC resources in the Experiences/Lessons Learned.

Round 2

Reviewer 1 Report

This is a much improved version. Still some issues such as for example not clear what metadata standards you are using and if the users are making use of them as well. Since this is not a publication interface (from what I can understand), the case of finding data or exchanging data  is not clear. Are users using the metadata to find their own  data or other users data? What does FAIR mean in  this context?  Also, not clear what is different from the current Cyverse Arizona. The audience for this paper is clearly an HPC audience, not sure that bio researchers will read the paper and think this is something for them. In the case of this general implementation papers it is necessary to focus the content and ask who are we talking to and what are the points we want to highlight? I hope  that this general comments enable you to get a publication out that sparks interest in an intended audience.

Author Response

Response to Reviewer 1

This is a much improved version. Still some issues such as for example not clear what metadata standards you are using and if the users are making use of them as well.

Response: We thank the reviewer for acknowledging the improvement of the publication. We added information on metadata standards (Dublin Core, MIGS, MIMS,...) as well as the usage in line 93-97. Some of our users make use of the metadata templates, however, some of them do also establish their own templates.

Since this is not a publication interface (from what I can understand), the case of finding data or exchanging data  is not clear. Are users using the metadata to find their own  data or other users data? What does FAIR mean in  this context? 

Response: We added the corresponding statement in line 89-90. CyVerse enables users to share data with collaboration partners. Then those partners will have in their archive a folder 'Shared with me'. So in principle, CAT users can search their own (meta)data as well as all (meta)data shared with them. So data will not be published via CAT, but it will be findable and accessible for collaborators.

Also, not clear what is different from the current Cyverse Arizona.

Response: We elaborated on differences between CAT and CyVerse US in line 109-118. (not accessible from outside, distributed storage, SGE connection, less modules).  

The audience for this paper is clearly an HPC audience, not sure that bio researchers will read the paper and think this is something for them. In the case of this general implementation papers it is necessary to focus the content and ask who are we talking to and what are the points we want to highlight? I hope  that this general comments enable you to get a publication out that sparks interest in an intended audience."

Response: We addressed this point in line 125 – 131. This publication provides an overview about a local infrastructure in Graz, Austria. It gives a detailed description of the technical setup at the universities in Graz, as well as an outlook for further developments which is in particular interesting for the HPC community. However, we think that LS researchers are looking more and more for new or already established solutions for RDM and reproducible analytics because publishers and funders are asking for FAIR solutions. Therefore, we assume that it is also a useful resource for them to understand the setup for a sophisticated RDM solution. And with the use cases we help them to identify as part of the target group. In addition, this will be also further investigated in our team in the course of a PhD thesis.